# *Origanum syriacum* L. (Za’atar), from Raw to Go: A Review

**DOI:** 10.3390/plants10051001

**Published:** 2021-05-17

**Authors:** Reem Abu Alwafa, Samer Mudalal, Gianluigi Mauriello

**Affiliations:** 1Department of Agricultural Science, University of Naples Federico II, 80049 Naples, Italy; abualwafa.reem@gmail.com; 2Department of Nutrition and Food Technology, Faculty of Agriculture and Veterinary Medicine, An-Najah National University, Nablus P.O. Box 7, Palestine; samer.mudalal@najah.edu

**Keywords:** *Origanum syriacum*, green thyme, essential oils

## Abstract

The interest in za’atar has increased in recent years due to its economic, cultural, and functional importance. The traditional za’atar mix made from dried *Origanum syriacum* is now a demanded product nationally and internationally. Air-drying at low temperatures can preserve za’atar quality traits better than other techniques such as oven-drying. The Palestinian za’atar market has the potential to develop and increase its value. However, it is facing many challenges. Another valuable product of za’atar is essential oil. Za’atar essential oil quantity and quality are affected by many factors including geographical location, cultivation, harvesting season, soil, extraction method, temperature, and others. These factors interact with za’atar and with each other; therefore, some factors are more effective than others and further research is needed to determine the optimum condition for producing and obtaining za’atar essential oil. Antimicrobial and antioxidant activities are the main functionalities of za’atar essential oil that are behind its medicinal importance. One hundred and twenty-one compounds have been identified in za’atar essential oil. The most common compounds are thymol, γ–terpinene, carvacrol, and α-pinene. The variation in the composition among za’atar essential oil samples results from the different conditions of the studies during za’atar growth as well as essential oil extraction.

## 1. Introduction

The word *Origanum* means “ornament of the mountains”, which comes from the Greek words oros that means mountain or hill, and ganos that means ornament. Since ancient times, *Origanum* species have been used as medicinal herbs. However, nowadays the importance of *Origanum* species in culinary practices has outgrown their therapeutic importance. Recently, several species have been used as ornamentals. The genus *Origanum* was first described by Linnaeus in 1754 [1], and it belongs to the Mentheae tribe Labiatae family [2]. More than 300 scientific names were given to less than 70 *Origanum* hybrids, varieties, species, and sub-species in the past 150 years. Seventeen hybrids, 6 sub-species, 38 species, 10 sections, and 3 groups were identified in 1980 by Ietswaart [1] based on the diverse morphological characteristics such as: stems length, stems and leaves indumentum, sessile glands number, verticillasters arrangement, and branches number, arrangement, and length. After that, one more hybrid and five more species were recognized [3]. Mainly, *Origanum* species are distributed in the Mediterranean region. Out of the 43 species, 35 species occur solely in the East Mediterranean [3]. *Origanum* species grow spontaneously in the Mediterranean region and get harvested once or twice per year during the flowering stage from wild population plants [3]. *Origanum* species can be propagated using seeds, cuttings, layering, or division. The species’ growing habit and variability will determine the best method for its propagation. The species *Origanum syriacum* is known by different names, such as “za’atar”, Lebanese oregano, Syrian oregano, and Bible hyssop [4]. It is a native crop in the eastern Mediterranean, western Asia, and southern Europe even though it has been cultivated in many parts of the world [5]. *O. syriacum* is a perennial herb with height ranging from 60 to 90 cm, woody creeping roots, and branched woody hairy stems [6]. Locally, the term za’atar (more correctly “za’atar mix”) is used to refer to designated herb mixes, in which traditional ingredients are added to the za’atar herb [7]. Za’atar flowers and leaves are commonly consumed after drying, grinding, and mixing with other ingredients. The strong and pleasant aroma of za’atar is related to its rich essential oil content [8]. The low cost, pleasant flavor, and diversity of *O. syriacum* culinary applications make it one of the most important stable herbs in the Arab region [9]. Za’atar products can be classified into three categories: fresh za’atar, dried za’atar, and essential oil herb [7]. Codex Alimentarius and ISO have established the standards for these products [10,11]. Za’atar production practices include seedling propagation, cultivation in the field, weed and pest control, irrigation and fertigation, and finally, harvesting and processing herb [7]. Limited data about processing effects on za’atar quality attributes are available. Hence, more studies are required to fully understand how za’atar is affected by processing and which processing condition can produce za’atar with optimum quality. Palestinian za’atar has a great economic value but many challenges are facing the Palestinian za’atar market growth [12]. Za’atar is also a powerful symbol of the Palestinian culture and it is strongly connected to the Palestinian identity [13]. *Origanum* secondary metabolites are well studied in both essential oils and polyphenolic compounds terms [14]. The major components of *O. syriacum* essential oil are thymol, carvacrol, sabinene hydrate, γ-terpinene, and p-cymene. Many factors can affect plants’ essential oil, including genotype, geographical location, altitude, harvesting time, vegetative stage, plant part, soil, drying and storage conditions, extraction method, and others [15]. *O. syriacum* has been used in folk medicine, especially in Arab countries [14]. Za’atar plants were found to have valuable functionality including antimicrobial, antioxidant, and other interesting activities. Several studies have been conducted to determine the composition of za’atar essential oil. Understanding all the intrinsic and extrinsic factors that influence the essential oil composition is necessary to determine the required conditions for producing za’atar and za’atar essential oil with optimum characteristics.

In this paper we have included an overview on the za’atar plant *O. syriacum* taxonomy, morphology, and distribution. We have also collected available data regarding za’atar cultivation, processing, and production. Finally, we have provided information about environmental and processing conditions that can influence za’atar essential oil, its functionality, and its composition.

## 2. Taxonomy of Genus *Origanum*

### 2.1. Origanum Morphology

*Origanum* can be generally characterized as several erect sub-shrubby Labiatae, and its stems are medium-sized, it has glandular punctate, and ovate leaves. *Origanum* sub-terranean parts are mostly woody. Many species of *Origanum,* especially the species grown in the East Mediterranean region, have woody, very thick roots. The stems in almost all Mediterranean species are woody as well. The length of stems varies among *Origanum* species but most of them have stems with a length range of 30 to 60 cm. The leaves of *Origanum* are petiolate in many species, and sessile or sub-sessile in several species. There is a big variation in the shape and size of leaf blades in *Origanum*. Leaves’ length ranges from 2 to 40 mm, and their width ranges from 2 to 30 mm. The indumentum of leaves is similar to stems; however, the hairs on leaves are usually shorter. Two types of glands exist on the leaves: sessile and stalked glands. These also found on other parts of the *Origanum* plant. Stems, corollas, bracts, and calyces also have glands on them. *Origanum* verticillasters are arranged in dense spikes and have few sessile flowers and differentiated bracts. The inflorescences are mostly paniculate, the spikes are usually compact, and the bracts are imbricate. The number of bract pairs in a spike ranges from 2 to 40 with a mean of seven. The shape of bracts is often roundish, ovate, or obovate. Small bracts are similar to leaves in their texture, indumentum, and color, while large bracts are papery, partly purple or a yellowish-green color, and often are not hairy or just slightly hairy. Only two sessile flowers are found on a verticillaster in most *Origanum* species. The nutlets of *Origanum* species that have been seen are not different. They are brown and ovoid with 1–1.5 mm length, 0.5 mm width, and a base that is slightly acute to apiculate. Pollen grains of some species have been studied. They were suboblate, hexacoblate, or oblatespheroidal [1,2,3].

### 2.2. Origanum Distribution

Mainly, *Origanum* species are distributed in the Mediterranean region. Out of the 43 species, 35 species occur solely in the east Mediterranean. Four species can only be found in the western Mediterranean, while three species grow in Libya exclusively [3]. Three taxa grow in Morocco and the south of Spain restrictedly, two taxa are found in Algeria and Tunisia, three taxa are endemic to Cyrenaica, and nine taxa are found in Greece, South Balkans, and Asia Minor. Six out of these nine taxa are local Greek endemics. Twenty-one local Turkish endemics are found in Turkey, Syria, Lebanon, and Cyprus. Eight taxa are distributed locally in Israel, Jordan, and Sinai Peninsula [2]. *O. vulgare* is the only species that grows in a very large area extended from the Azores to Taiwan, while most of the other species occupy small areas. A total of 70% of *Origanum* groups are endemic to a single island or mountain [1]. Natural and artificial hybrids of *Origanum* are found when *Origanum* species co-occur. Usually, hybrids are considered as species initially [3]. Despite that *Origanum* hybridization is common, hybrids are not usually found in large numbers. Hybridization is hypothesized to be the main method for the origin of *Origanum* species. It may be achieved between two species of *Origanum* or between an *Origanum* species and a related genera species [1]. The most widely distributed hybrid is O. x intercedens Rechinger, which is found in large populations in the Aegean islands [2]. Most *Origanum* species grow in mountains and headland areas. They often grow in open or mixed coniferous woods in a partially shaded environment. However, some species, especially those belonging to the Majorana section, are found in lower altitude levels. *Origanum* species mostly grow on calcareous soils and almost all of them can grow on stony slopes, rocky places, and cliffs [1].

## 3. Production of Za’atar Mix

### 3.1. Za’atar Products

Locally, the term za’atar is used to refer to designated herb mixes, in which traditional ingredients are added to the za’atar herb [7]. Za’atar flowers and leaves are commonly consumed after drying, grinding, and mixing with other ingredients [8], such as seeds, sesame, sumac berries (*Rhus coriaria*), and salt [4]. This mixture is served for breakfast [8] in combination with olive oil spread on a pizza-like pastry called “Man’ousheh” in Arabic [7], and it can also be used in cooking for seasoning [8]. Za’atar mix also became a popular filling for French croissants. For za’atar’s known therapeutic benefits, it has been consumed as herbal tea in combination with other herbs despite its sharp and bitter flavor compared to other types of herbal tea [7]. The strong and pleasant aroma of za’atar is related to its rich essential oil content [8]. The low cost, pleasant flavor, and diversity of *O. syriacum* culinary applications make it one of the most important stable herbs in the Arab region [9]. Za’atar products can be classified into three categories: fresh za’atar, dried za’atar, and essential oil herb. The consumer has higher acceptability of fresh za’atar in autumn and winter for its milder taste due to the reduction in the essential oil concentration caused by rainfall. Nevertheless, the harvesting of fresh za’atar must end in early spring, which is the pre-blooming period, in order to give time for the blooming biomass to recover before the harvesting time of dried za’atar in July and June. Dried Za’atar mix products are composed of leaves and inflorescences. Their final form is grounded pure za’atar or za’atar with other ingredient mixes. The harvesting of za’atar for dried za’atar products is preferred to be carried out during the full blooming stage. Essential oil products are produced from the leaves and inflorescences of za’atar in an extracted essential oil or water form. The market value of different essential oil profiles is different. Therefore, knowing and testing the market need is necessary in order to produce the requested profile [7].

### 3.2. Za’atar Mix Composition

According to the Codex Alimentarius Commission Food and Agriculture Organization—World Health Organization, FAO–WHO, 2017 on za’atar mix standardization, there are three types of za’atar products: raw za’atar, raw broadleaf za’atar, and mixed za’atar. Raw za’atar products are composed of the leaves and/or blossoms of the wild and cultivated plant species: *Origanum*, *Thymbra*, *Thymus*, and *Satureja*. They could be mechanically or manually crumbled but they are not provided in a powder form. Raw broadleaf za’atar products are made of the leaves and/or blossoms of wild or cultivated broadleaf za’atar from the species *O. syriacum* (by at least 75%) or as a constitute (by 25% maximum) in a mixture of leaves and blossoms of the varieties: *Origanum ehrenbergii*, *Thymbra spicata*, *Coridothymus capitatus*, *Thymus syriacus*, and *Satureia thymbra*. Like raw za’atar, the mix could be manually or mechanically crumbled but it is not provided in a powder form. Mixed za’atar products are made of raw za’atar and raw broadleaf za’atar in addition to the husk sumac and sesame seeds. Other ingredients could also be added. Za’atar mix products are classified into three categories: Premium, Extra, and Regular mixed za’atar. Premium mixed za’atar must contain at least 25% broadleaf za’atar mixed with sumac husk and sesame seeds only. A maximum of 6% salt can be added. Extra mixed za’atar contains 20% broadleaf za’atar and raw za’atar mixed with sumac husk and sesame seeds. Grains, nuts, spices, and condiments could be added as well. Like Premium mixed za’atar, only a maximum of 6% salt can be added. Regular mixed za’atar composition is at least 15% broadleaf za’atar and raw za’atar mixed with at least 5% sumac husk and sesame seeds in addition to other ingredients. Regular za’atar mix can contain aromatic grains and herbs, legumes, pomegranate molasses, vegetable oil, wheat bran, sesame seed hull, spices, condiments, and nuts, all of which must meet good manufacturing practices. It is possible to add a maximum of 7% salt and 4% citric acid with an indication of that on the label [10].

### 3.3. Quality Factors for Za’atar Mix Products

Za’atar must have special flavor and odor without any external flavors and odors that may come from molds, rancidity, or any other substances. Additionally, the color of za’atar mix products should be normal and consistent with the color of such products. All ingredients used for the preparation of za’atar mix products should meet their Codex Alimentarius standards. They should not contain any living insects or spiders, and should be reasonably free from any noticeable molds, dead insects, parts of insects, or rodents, birds, and snail waste. Za’atar mix products must not be in a powder form to guarantee that all the main ingredients of the mix can be seen at a microscopic level or by the naked eye. This will ensure avoiding frauds and impurities concealment. It will also maintain a higher level of essential oil. If the product contained straws, they should not make more than 5% of the mass of the products, and they should not be more than 10 mm in length and 2 mm in diameter. Any non-vegetable origin substance such as soil, dust, sand, etc., or non-food vegetable substance such as wood or dry leaves found in za’atar mix products, should make more than 1% of the mass of the product. Table 1 shows Codex Alimentarius specific chemical composition requirements for each za’atar mix category products. Only regular za’atar mix can contain the additive citric acid in a quantity that does not exceed 4 mg/kg [10]. The International Organization for Standardization (ISO) established a specification for whole or ground-dried *Origanum* species and sub-species leaves in both processed and semi-processed products, and published it in the ISO 7925 report in 1999. The ISO requirements included: oregano odor and flavor must be aromatic, warm, fragrant, pungent, and bitter, and be free of foreign flavors; oregano products should be free from insects, molds, etc.; extraction matter should not exceed 1% by mass in processed products and 3% in semi-processed products; and oregano plant parts such as stalks should not exceed 3% in the products. Ground oregano must completely pass through a sieve with a 500 μm nominal aperture size. Table 2 shows the ISO chemical requirements for dried oregano. Oregano product packs should be sound, dry, clean, and made from materials that protect the product against foreign bodies, loss or ingress of moisture, and volatiles loss without affecting it. The label of the product must include the following: product name, type, trade name, producer or packer address, trademark, batch code or number, net mass, produced country, any other consumer-required information such as harvesting year and packing date, any treatments of the product, and the reference to the ISO standard [11].

### 3.4. Za’atar Mix Production Process

Za’atar production practices include seedling propagation, cultivation in the field, weed and pest control, irrigation and fertigation, and finally, harvesting and processing. Za’atar has a very short shelf life that is only 2 days. Therefore, they must be stored away from heat and dry wind in order to keep their freshness [16]. Za’atar must be dried soon after harvesting. Za’atar is dried in shade and well-aerated areas. Harvested za’atar must be handled in a proper way and stored in suitable clean bags and containers that are aerated. Drying should be carried out in a shaded, well-aerated area isolated from direct sunlight and any pollution source. Because harvesting for dried za’atar takes place in summer, the drying process may last from 2 to 3 weeks depending on the sunlight intensity and relative humidity in the drying area [16,17]. The temperature in the drying area should not be higher than 30–35 °C to prevent the loss of essential oil by evaporating. Once drying of za’atar is achieved, it can be stored for several months if the storage area is dry and isolated from moisture, sunlight, heat, and polluting substances. Dried za’atar can be manually or mechanically threshed [18]. Za’atar grinding is the next step after threshing and sieving dried flowers and leaves to further mix with other ingredients to produce the za’atar mix. Final storage of raw dried, threshed, and ground za’atar must be in hygienic conditions. Lastly, dried za’atar should be packed in clean, suitable packages made from materials that protect the product from moisture and prevent the loss of its volatiles without affecting it [7].

### 3.5. Effect of Processing on Quality Traits

Limited data about the processing effects on za’atar quality attributes are available. In Table 3 and Table 4 we report the main effects on the product of pre- and post-harvesting conditions, respectively. The different stages in za’atar processing can influence its quality in different ways. It was found in a study that medium irrigation was the most proper irrigation level for *O. syriacum*. It also reported that air-drying was the best for preserving the aroma, flavor, and color compared to oven- and freeze-drying [16]. Air-drying was also reported to be more favorable for preserving total phenols (TP) and antioxidant activity in herbs, including oregano, in a study compared to vacuum oven- and freeze-drying [17]. Another study found that drying at a lower temperature was better for preserving chlorophyll and carotenoids contents in *O. syriacum,* while drying at a higher temperature produced *O. syriacum* with higher phenols content. It was also reported that shade drying at lower air velocity was the best for preserving alpha-tocopherol content [18]. Every processing method has its strengths and weaknesses, and every quality trait in za’atar responds differently to processing conditions. Therefore, more studies are required to fully understand how za’atar is affected by processing and which processing conditions can produce za’atar with optimum quality. Atallah et al. [16] investigated the effect of different irrigation, drying, and production methods on the postharvest quality and economic feasibility of *O. syriacum* growing in Lebanon. The resulting data collected during two summers showed that *O. syriacum* height significantly and positively increased with irrigation frequency level from low to medium (66.3–73.2 cm) and from medium to high (73.2–81.5 cm). Similarly, branching increased with increasing irrigation level, reaching the maximum branching with medium irrigation level, and starting to reduce slowly beyond this level. Accumulation of dry matter as well was negatively affected by increased soil moisture. Therefore, it was suggested that medium irrigation water stress level is required for the prevention of reduced plant branching and dry matter accumulation. An increase of 1.7- and 2.2-fold was observed when increasing the irrigation level from low to medium. However, additional increases beyond medium level did not show more increases. Flower yield was increased by 2-fold in the high irrigation level but the validity of this conclusion is limited due to the high standard deviation of the flower yield data. The levels of thymol and carvacrol in the essential oil of *O. syriacum* were not statistically different in the different irrigation levels. Drying of *O. syriacum* was carried out by three different drying technologies: air-drying in a shaded area for 10 days at 25 °C and 45% relative humidity, oven-drying by convection oven at 30 °C for 47 h in 1.2 m/s air velocity and freeze-drying at 0 °C initial temperature for 48 h. The results of the sensory analysis of dried *O. syriacum* revealed that air-dried *O. syriacum* had a higher preservation of aroma and flavor than the oven- and freeze-dried *O. syriacum*. The flavor of air-dried *O. syriacum* was marked as strong, bitter, pungent, warm, and astringent, while the aroma was marked as camphor-like, strong, and pungent. Freeze-dried *O. syriacum* flavor and aroma received lower scores with increased undesired earthy and musty aroma. This could have resulted from the presence of sesquiterpenes in freeze-dried *O. syriacum*. Both oven- and freeze-drying would cause a loss of carvacrol in the essential oil of *O. syriacum,* leading to the loss of the warm and pungent aroma and flavor. There were significant differences among color scores of air-, oven-, and freeze-dried *O. syriacum.* According to the study experts, air-drying produced the desirable grayish-green color that is also required by the international standards. Financial analysis showed that the medium irrigation level and air-drying production scenario was not found to be economically feasible due to the high operation and maintenance costs, mainly. However, reducing the product positioning, e.g., by using low-value jars, might make this scenario economically feasible.

Hossain et al. [17] investigated the effect of air-, freeze-, and vacuum oven-drying on TP, rosmarinic acid, and antioxidant capacity in six Lamiaceae herbs: oregano, rosemary, marjoram, thyme, sage, and basil. All were harvested from the northern Negev Desert in Israel. Antioxidant capacity was measured by ferric-reducing antioxidant property (FRAP) and oxygen radical absorbance capacity (ORAC). Air-drying was carried out in a dark and well-ventilated area for 3 weeks at room temperature with a mean of 14 °C and 10% relative humidity. On the other side, vacuum oven-drying was carried out at 70 °C for 16 h, while freeze-drying was carried out at −54 °C for 72 h and further water sublimation under vacuum. A wide variation was observed among the content of TP, FRAP, and ORAC values. The lowest TP content range was found in a fresh sample of sage on day 0, while the highest TP content was found in the air-dried sample of thyme at day 0. Similarly, the fresh sage sample at day 0 had the lowest FRAP value, and air-dried thyme at day 0 had the highest FRAP value. Air-dried thyme also had the highest ORAC; however, the lowest ORAC was found in fresh rosemary on day 0. No significant (*p* < 0.05) effect of storage on the assays was observed, except in the fresh samples. This could have resulted from the immediate vacuum-packed at −20 °C storage after the treatments. The absence of oxygen together with the low temperature in the storage environment inhibited the enzymatic oxidation of antioxidants in dried herbs. On the other hand, storing fresh samples at −20 °C caused severe chilling injury due to the presence of a high level of moisture and the loss of plants’ integrity. The highest release of antioxidants in fresh samples was observed on day 15 with no further loss observed after that. However, the woody herb thyme was not affected by the chilling injury and no increase in its TP and antioxidants capacity occurred in the fresh thyme sample when stored at −20 °C. All three drying methods were similar in their efficiency and produced typical amounts of dry weight. The higher contents of TP in the dried samples compared to the fresh ones were suggested to be caused by the breaking of cell walls when the brittle dry herbs were milled and homogenized, which increased the release of phenolic compounds. Another suggested reason was the enzymatic degradation of antioxidants in the fresh sample, where the enzymes were active due to the presence of moisture. Air-dried samples had the highest levels of both TP content and antioxidant capacity. In air-drying, herbs lose moisture gradually, which might allow the metabolically active plants to sense this stress and produce phenolic compounds as a defense mechanism. This was proved by the significantly (*p* < 0.05) higher content of rosmarinic acid in air-dried samples compared to both fresh and other dried samples. Rosemary and thyme were the only herbs that showed significant (*p* < 0.05) differences in TP content and antioxidant capacity among vacuum oven-dried and freeze-dried samples. The vacuum oven-dried rosemary sample showed significantly higher TP content and FRAP level than freeze-dried rosemary, but no significant ORAC difference was found between them. Thyme showed the highest difference between TP content and antioxidant capacity in the vacuum oven- and freeze-dried samples among all herbs. Intact tissues in woody herbs such as thyme and rosemary act as barriers against phenolic compound release, which reduces their antioxidant capacity. The higher levels of TP content and antioxidant capacity in vacuum oven-dried samples than freeze-dried samples might be due to the reactivation of degradation enzymes in freeze-dried herbs or the release of phenolic compounds by the high heating temperature during vacuum oven-drying. Wakim et al. [18] studied the influence of various drying conditions on the quality of two types of Lebanese *O. syriacum* from Rkai and Ibrine regions. Solar drying, shade drying, and four drying treatments using forced air artificial dryers with two different temperatures (26 and 45 °C) and two different air speeds (0.1 and 0.2 m/s) were used. For both *O. syriacum* types, the decrease in chlorophyll content was more noticeable at the high temperature of the solar drying and the dryers at 45 °C. On the other hand, there was not a noticeable loss in chlorophyll content at 26 °C with 0.2 m/s drying, followed by shade-drying, and then 26 °C with 0.1 m/s drying. It was reported that pheophytinization accounted for the loss of chlorophyll. There was a similar rate of thymol and carvacrol loss in both *O. syriacum* types, and drying at 45 °C caused a lower loss than other drying treatments. The loss of polyphenols was found to be the highest after drying at 26 °C and 0.1 m/s and lowest after solar-drying and drying at 45 °C and 0.2 m/s. This might have resulted from the decomposition of some phenolic compounds by heat. This action produces other structures that increase the level of phenolic compounds after drying at a high temperature. Besides, drying at lower temperatures in shade-drying and drying at 26 °C showed a significant loss in polyphenols due to the prolonged time of drying. Shade-drying was the best method for preserving alpha-tocopherol in both *O. syriacum* types. A great loss in all tocopherols was caused by drying at a high temperature. Moreover, drying at 0.1 m/s air speed was more advantageous than drying at 0.2 m/s. Higher losses of tocopherols in Ibrine type than Rkai type were observed after drying at 45 °C and 0.2 m/s. Similarly, the tocopherol content loss was also lower in Rkai type after drying at 45 °C and 0.1 m/s. It was suggested that those differences are due to the presence of iron or copper sulfate in Ibrine type, which can speed up tocopherol oxidation. Drying in shade is the most proper method to preserve tocopherols in both types of *O. syriacum,* because the high temperature that increases the loss of tocopherols is not used in this method. Carotenoid loss rates were higher after drying at the higher temperatures in solar-drying and at 45 °C than drying at 26 °C. The loss of carotenoids was more pronounced in Rkai type. Carotenoids are easily decomposed in the presence of oxygen, light, and copper, and by the effect of temperature. Thus, solar-drying and drying at 45 °C resulted in higher losses of carotenoids than shade-drying and drying at 26 °C. Another reason behind the higher preservation of carotenoids in drying at a lower temperature is the lower oxidation of fat that produces peroxides that can destroy carotenoids. It was difficult to determine a favorable method of drying for the preservation of all the substances in the study.

### 3.6. Palestinian Market of Za’atar

Palestinian za’atar (*O. syriacum*) is not just a symbol of Palestine; it has also a great economic value. The economic value of growing and processing za’atar had become more viable in the last decade. The za’atar market has been greatly growing in recent years. Over 6000 dunams are planted with *O. syriacum* in the northern West Bank, while only 500 dunams were planted in za’atar before 2010. The increasing demand by foreign companies has expanded the cultivated land of this herb. The za’atar market has a good profit; it regains its capital quickly, keeps producing for 7 years, has an average of harvesting every 40 days, and is resistant to pests and diseases. Palestinian za’atar farmers earn about 144 million New Israeli Shekel (NIS) annually. More than 85% of processed za’atar is sold in the local market, and the other 15% goes to the export market. The majority of produced za’atar is fresh-cut za’atar with an estimated 15,000 tons annually. The final marketed product of za’atar ranges from 20 to 29% (average 24%) compared with the freshly harvested biomass. Commercial production of za’atar is new in Palestine; it represented only 3% of the total exported herbs in 2013, but the production is growing and it more than doubled in both value and volume after two years, reaching 763 tons of dried za’atar and 329 tons of fresh za’atar in December 2016, according to the Palestinian Ministry of Agriculture. However, there is an export demand for other types of za’atar that are not known locally, but there are no research centers that can work on the hybridization and development of such kinds of za’atar. Palestinian *O. syriacum* is not labeled by its scientific name; it is usually labeled as oregano or its Arabic name, za’atar. The end market of Palestinian za’atar is still dominated by about 42 cooperatives, hundreds of women’s groups, small-scale farmers, and about 16 processing industries. These processers are mainly located in Nablus and Jenin where most za’atar harvesting is carried out. A total of 200 workers produce za’atar and other kinds of herbs in these processers. A total of 70% of za’atar production in Palestine is done by commercial manufactures, while the other 30% is produced at homes and consumed locally. Some of the manufacturers buy fresh za’atar from farmers in big volumes during the peak of the season to reduce the costs. In Palestine, za’atar is commonly consumed in a dried form either alone or in za’atar mixes. Palestinians and people in regional countries prefer za’atar mixes for their strong, unique flavor. Only a small amount of za’atar from Jordan and Turkey is imported to Palestine each year, while Palestine exports za’atar to many countries including Jordan, Canada, Gulf countries, EU, and USA countries. About 70% of exported za’atar is in the dried form and most of it is exported in bulk to Jordan. Only some private-labeled packaged za’atar products are exported to Europe and the US. The Palestinian za’atar market is not big enough to reinforce an active industry. Therefore, za’atar producers and processors should concentrate on the export market to make a high-quality volume. To achieve this, local producers and processors will need to invest in developing export capabilities. Moving from exporting bulk raw za’atar to packaged branded products will significantly increase the Palestinian export value. Unfortunately, many challenges are facing the Palestinian za’atar market’s growth. It was found that Palestinian za’atar is re-exported from Israel to important markets as Israeli za’atar. Additionally, the cultivation of za’atar in Palestine is still difficult since its native areas are under occupation. Another challenge is the presence of competitor suppliers such as Jordan, Turkey, Lebanon, and Syria. These suppliers can compete on price, quality, and consistency of supply. Palestinian growers face difficulty in providing products with consistent quality, supply, and price. Manufacturers prefer other sources to buy raw materials, especially sumac and sesame, where they buy it from Turkey, Ethiopia, and Egypt. Other problems in Palestinian za’atar processing and production include: farmers are unorganized, the use of unsafe fertilizers, lack of equipment, lack of safety and standards certifications, some essential ingredients’ availability, quality and high cost, and very little production and processing of organic za’atar. Many factors need to be improved to solve these problems, including agriculture and propagation practices, research, processing and preservation technologies, quality control, marketing, legalization, and others [12].

## 4. *Origanum syriacum* Essential Oil

Several studies have been conducted to determine the composition of za’atar essential oil. One hundred and seven compounds have been identified in 19 za’atar essential oil samples in 13 studies conducted on za’atar from Syria [26], Turkey [5,29,30], Lebanon [6,28,31,32], Egypt [33,34,35], Israel [33], Côte D’Ivoire [36], and Palestine [19]. The levels of compounds were reported as their chromatogram areas’ percentages. One hundred and seven different compounds were found in za’atar samples. The most common compound was thymol, which was detected in all essential oils, followed by γ–terpinene and carvacrol, which were found in 18 essential oils, then α-pinene, found in 17 samples, and α-terpinene, found in 16 samples. Other common compounds were p-cymene, β-pinene, caryophyllene oxide, α-terpineol, α-thujene, terpinen-4-ol, camphene, and borneol. Among all compounds in za’atar essential oils, carvacrol, thymol, α-terpinene, o-thymol, γ–terpinene, ß-myrcene, cis-sabinene hydrate, terpinen-4-ol, p-cymene, caryophyllene, and β-caryophyllene had the highest levels. Compounds with the highest variations among their levels in the studies were carvacrol, thymol, ß-myrcene, α-terpinene, cis-sabinene hydrate, γ–terpinene, and β-caryophyllene. There was a wide variety of the detected compounds as well as their levels in the za’atar essential oils. All studies extracted essential oil from air- or shade-dried za’atar, except for Fleisher and Fleisher [34], Loizzo et al. [31], and Viuda-Martos et al. [35], where essential oils were obtained from fresh plants. Eight studies used hydro-distillation for essential oil extraction [6,28,29,31,32,34,35,36], while four studies used steam distillation [5,26,30,33], and one study used a microwave-ultrasonic method to extract the essential oil [19]. All studies used GC-MS for the analyses of essential oil, except for Al-Mariri et al. [26], where they used GC-HPLC. Kouame et al. [36] used Carbon-13 nuclear magnetic resonance (13C-NMR) spectrometry besides GC-MS. Quality control is a major challenge in herbal products and supplements [37]. In addition to the difference in experimental conditions in obtaining and analyzing za’atar essential oil in studies, the environmental and processing conditions can also affect za’atar essential oil in both quality and quantity aspects.

### 4.1. Geographical Location and Cultivation

The harvesting location and cultivation method have interesting effects on the yield and composition of *O. syriacum* essential oil. These effects were noticed in six studies conducted in different countries [19,20,21,22,23,33], while only one study did not report an effect of the harvesting location on za’atar essential oil composition [38]. A difference between the levels of geraniol and geranyl esters in za’atar essential oil from Egypt and Israel was observed in one study [33], while five studies reported differences in *O. syriacum* essential oil yield and its main components thymol and carvacrol levels in different geographical locations and cultivation methods [19,20,21,22,23]. However, the observations of these studies are contrary. In two studies it was reported that the higher altitude produced *O. syriacum* with the higher essential oil yield [21,23], while the opposite was reported in the other two studies [19,22]. For the essential oil composition, carvacrol was the main compound in essential oils obtained from *O. syriacum* harvested from the higher altitude locations, and thymol was the main compound in essential oils obtained from za’atar harvested from the lower altitude locations in three studies [21,22,23]. In one study, carvacrol and thymol were the main compounds in essential oils obtained from *O. syriacum* harvested from the lower altitude locations, and γ-terpinene and α-terpinene were the main compounds in essential oils obtained from *O. syriacum* harvested from the higher altitude locations [19]. It is important to note that thymol was the main compound in wild *O. syriacum* essential oil, and carvacrol was the main compound in cultivated *O. syriacum* essential oil in one study [20], but in another study, when wild *O. syriacum* was harvested from a high-altitude location it had carvacrol as the main component, and cultivated *O. syriacum* harvested from a low-altitude location had thymol as the main component [22]. We suggest that although all environmental factors can affect za’atar essential oil, some factors such as altitude have a stronger effect than other factors, such as cultivation.

#### 4.1.1. Harvesting Season

Harvesting time has an obvious effect on *O. syriacum*’s essential oil yield. On the other hand, harvesting time effect on *O. syriacum* essential oil composition is variable. Four studies reported that the highest essential oil yields were obtained when *O. syriacum* was harvested during the summer [20,21,23,24]. For *O. syriacum* essential oil composition variation, both thymol and carvacrol were reported to be higher in summer during the flowering period in two studies [21,23]. Another study found that wild *O. syriacum* leaves produced a higher level of thymol and cultivated *O. syriacum* leaves produced a higher level of carvacrol in June after flowering [25]. Thymol level was observed to be increased in May in one study [20], while in two studies thymol levels were higher when *O. syriacum* was harvested in early spring before flowering and during winter and autumn, and the levels of carvacrol were higher when *O. syriacum* was harvested in spring and summer [8,24]. For γ-terpinene and p-cymene levels, two studies reported that they were higher when *O. syriacum* was harvested in spring between February and May [20,25], while one study reported that p-cymene and thymoquinone levels were higher after the flowering stage from January to July [23]. It was found in one study that the level of myrsitic acid was the highest after flowering in June, and for α-linolenic acid and palmitic acid, levels were the highest in April before flowering. Additionally, the mineral content obtained from cultivated *O. syriacum* was higher in April after flowering [25].

#### 4.1.2. Soil

Soil pH and composition have an impact on both *O. syriacum* essential oil yield and composition. Different soil types and conditions have been found to affect *O. syriacum* essential oil in two studies. Choosing the proper soil type and conditions will depend on the compound of interest in the essential oil and how they affect it [15,23]. There was a significant positive correlation between essential oil yield and soil pH as well as soil phosphorus content (*p* < 0.05), while the positive correlation between essential oil yield and organic matter, nitrogen, and potassium was low and insignificant. Gas chromatography-mass spectrometry (GC-MS) results show that soil pH and composition affected the essential oil contents. Both soil pH and P_2_O_5_ content were significantly negatively correlated with carvacrol, thymoquinone, and p-cymene, and they were significantly positively correlated with thymol, while thymol was significantly negatively correlated with nitrogen, K_2_O, and organic matter content [23]. For soil types, manure, vegetable compost, non-sterilized, and non-inoculated soils produced *O. syriacum* essential oil with high thymol content. On the other hand, nursery, potting, professional agricultural mixtures, sterilized inoculated, sterilized non-inoculated, and non-sterilized inoculated soils produced *O. syriacum* essential oil high in both thymol and carvacrol. Water, organic matter, and mineral rate improve the essential oil yield. Moreover, arbuscular mycorrhizal fungi (AMF) enhances the water and nutrient intake and plant growth, which improves essential oil yield as well. It was noted that AMF promoted the carvacrol and reduced the thymol levels in *O. syriacum* essential oil [15].

#### 4.1.3. Extraction Methods Effect on Za’atar Essential Oil

Besides the effect of the environmental factors on za’atar essential oil, extracting and identification methods can also have an impact on za’atar essential oil yield and composition. Hydro-distillation produced higher essential oil yield compared to supercritical-fluid-extraction (SFE) in a study [27]. One study reported that NaCl had increased *O. syriacum* essential oil yield before and during extracting [21]. Static headspace (HS) was found to have better preservation of volatiles compared to steam distillation (SD) in a study [20]. Another study found that solid-phase microextraction (SPME) had better preservation of monoterpene hydrocarbons, while these compounds changed into the oxygenated hydrocarbon monoterpene in GC-MS under the effect of heat and time [27]. It was reported in a study that hydro-distillation had better identification of essential oil components compared to solvent extraction. However, the levels of each identified component were enhanced in solvent extraction depending on their solubility in the different solvents [28].

#### 4.1.4. Other Factors

Other factors that were reported to influence *O. syriacum* essential oil yield included the plant part in which essential oil was extracted [21], the drying technology [23], and the level of drying [36], while using growth-enhancing bacteria was found to affect both *O. syriacum* essential oil yield and composition [39].

### 4.2. Origanum syriacum Essential Oil Functionality

#### 4.2.1. Medicinal Uses of *O. syriacum* and Its Essential Oil

The medical importance of the *Origanum* genus comes from its antioxidant, antimicrobial, antifungal, antispasmodic, antimutagenic, anti-inflammatory, and analgesic properties. It has been used as an ethnomedicine for thousands of years. *Origanum* has been used as carminative, stimulant, tonic, and diaphoretic. Turkish folk medicine has used *Origanum* for gastrointestinal problems as an antiparasitic, antihelminthic, and expectorant. It is suggested that the synthesis and release of inflammatory mediators are influenced by the carvacrol in *Origanum* essential oil. Thus, it makes it useful in treating gastric ulcers. Folk medicine also uses *Origanum* in the treatment of cough, colic, irregular menstrual cycle, and toothache. It has also a promising potential in preventing diabetes complications [14]. Many studies confirmed the beneficial health effects of *Origanum* and its different roles in the treatment of various health problems including respiratory, gastric, and urinary tract disorders, dermatological affections, viral infections, and cancer [3]. Besides its medical uses, *Origanum* can be used as a disinfectant and flavoring agent for soups, perfumes, and food products [14]. *Origanum* species that have been used for their medicinal properties include *O. syriacum*, *O. vulgare*, *O. dictamnus*, and *O. majorana* [4]. *O. syriacum* has been used in folk medicine, especially in Arab countries. It has an antiseptic activity, and it is helpful in stomach and intestinal pain relief, heart disease, cough, cold, wounds, toothache, and anxiety [14]. The famous Persian doctor and philosopher Ibn Sina (980–1037 AD) mentioned *O. syriacum* effects in his worldwide popular book “The Canon of Medicine”. He said: “it is a good analgesic for joint pain, chewing the leaves relieves both gum and toothache. When rubbed upon chest it relieves bronchitis, it has a beneficial effect on the liver and stomach and a strong anthelmintic effect”. Additionally, the doctor, pharmacist, and astronomist Dawud ibn ‘Umar al-Anṭākī (1534–1592 AD) talked about *O. syriacum* in one of his books. He said that it is “an antidote for many poisons, a carminative, for detoxification of the organism, as a blood thinner, for loss of appetite, as an anthelmintic and food preservative.” [6]. The uses of *O. syriacum* in folk medicine are different according to the country. For example, it is used for treating stomach pain in Jordan, while in Turkey it is used for cold symptoms mainly [32].

#### 4.2.2. Antimicrobial Activity of Za’atar Essential Oil

An abundant number of studies have reported the antimicrobial activities of *Origanum* essential oil and its components. These activities have been mainly classified as antibacterial and antifungal activities. It is believed that phenols are the most powerful antimicrobial compounds in the essential oil. The next most powerful antimicrobial compounds are alcohols, followed by ketones, ethers, and then hydrocarbons [3]. The antibacterial activity gives plants importance in treating infectious bacterial diseases. *O. syriacum* has an antibacterial property against both Gram-positive and Gram-negative bacteria, which make it useful for both bacterial infectious diseases and food preservation. Essential oils’ hydrophobic nature allows them to be embedded in the membranes of the cell and the mitochondria, break them, and throw out their contents. Some conditions such as low pH, temperature, and oxygen levels can improve its antibacterial activity. Similarly, antifungal activity is also an important property of *O. syriacum* essential oils. It plays a role in preserving food and treating infectious diseases, as well as plant pathogenic fungi that can cause significant losses for farmers [40]. Many microbes were found to be sensitive to *O. syriacum* essential oil at different degrees. Variable concentrations and concentration units were used by the studies. *O. syriacum* essential oil’s antimicrobial effect is mostly concentration-dependent. It has interesting inhibitory effects that are comparable to synthesized antimicrobial effects. It was also reported in a study that *O. syriacum* essential oil could improve the effect of some antimicrobials [41]. *O. syriacum* essential oil was effective against antimicrobial-resistant bacterial strains in two studies [29,41]. Some studies even found antimicrobial effects of the individual components in *O. syriacum* essential oil [19,26,32]. Different factors were found to affect the antimicrobial activity of *O. syriacum* essential oil. It was reported that the essential oils with the richest γ-terpinene and α-terpinene levels have the highest antibacterial activity in a study [19]. Cultivated *O. syriacum* essential oil was found to exhibit higher antimicrobial activity than wild *O. syriacum* essential oil in a study [22]. Table 5 shows the microorganisms that are affected by *O. syriacum*.

#### 4.2.3. Antioxidant Activity of Za’atar Essential Oil

Antioxidant dietary supplies from *Origanum* plants have been considered effective scavengers of the human body’s metabolic pathway free radicals [3]. A large number of studies found an antioxidant activity in *O. syriacum*. Antioxidant activity protects cells from harmful free radicals and reactive oxygen species (ROS). The antioxidant activity also improves endothelial function, stimulates DNA repair, and has anti-inflammatory action. Many studies indicated that phenolic compounds and flavonoids in essential oils are responsible for the antioxidant activity. It was also found that *O. syriacum* essential oil exhibits an antiparasitic activity that might be due to its phenolic compounds, such as thymol and carvacrol, flavonoids, and terpenoids [40]. Assessing the antioxidant activity of a product should be carried out using more than one technique. Using one technique can give information about the antioxidant activity of the sample, but combining different techniques will give more detailed information on the antioxidant activity. Comparing the values of antioxidant activity obtained in different laboratories is difficult, even if the same technique was used, due to the differences in sample preparation, antioxidants extraction, end-points selection, and results expression. Therefore, combining different techniques is necessary [35]. *O. syriacum* essential oil showed antioxidant activity in a wide range of assays. Similar to the antimicrobial activity, variable concentrations and concentration units were used by the studies. One study tested the antioxidant capacity of *O. syriacum* essential oil compounds as well [42]. Three studies reported that the antioxidant capacity of *O. syriacum* essential oil was concentration-dependent [5,19,45]. Another study found that dried *O. syriacum* essential oil exhibited higher antioxidant activity than fresh *O. syriacum* essential oil [36]. It was reported in a study that the essential oil with the richest γ-terpinene and α-terpinene contents had the highest antioxidant capacity [19]. Table 6 summarize the antioxidant assays used to measure the antioxidant activity of *O. syriacum*.

#### 4.2.4. Other Activities

Several interesting activities of za’atar essential oil were found, including tocolytic [24], anti-inflammatory [31,46], anticholinesterase [31], amoebicidal [47], cytotoxic [48], hypoglycemic, pancreatic enzymes inhibition [38], and insecticidal activities [30]. No adverse effect of *O. syriacum* oral administration on blood was found in an animal study [48], which makes it a promising natural therapeutic agent with many properties. However, animal and human studies are needed to approve its safety and effectiveness.

## Figures and Tables

**Table 1 plants-10-01001-t001:** Codex Alimentarius chemical requirements for za’atar [10].

Characteristics		Requirements		
		Premium Mixed	Extra Mixed	Regular Mixed
Moisture % (*w*/*w*) maximum		12	12	12
Total table salt % (*w*/*w*) maximum *		6	6	7
Total ash, excluding salt % (*w*/*w*) maximum *		7	7	7
Total ash % (*w*/*w*) maximum *		14	14	15
Acid insoluble ash % (*w*/*w*) maximum *		1	1	1
Raw fibers % (*w*/*w*) maximum *		16	15	37
Volatile oils % (*v*/*w*) minimum *		0.37	0.13	0.1
Maximum superoxide number		-	-	10 mL of superoxide oxygen/kg of oil
Malic/citric acid proportion minimum		10	10	0.14
Basic Components Volatile Oils	Carvacrol + Thymol	>70%	>85%	>85%
	Cymene, gamma-terpinene, and other volatile oils	<30%		

* on dry matter.

**Table 2 plants-10-01001-t002:** ISO Chemical Requirements for dried oregano [11].

Characteristic	Specification			Test Method
	Whole or Cut Leaves		Ground (Powdered)	
	Processed	Semi-Processed		
Moisture content % (by mass) maximum	12	12	12	ISO 939
Total ash % (by mass) on dry basis, maximum	10	12	12	ISO 928
Acid-insoluble ash % (in mass) on dry basis, maximum	2	2	2	ISO 930
Volatile oil content, mL/100 g on dry basis, minimum	1,8	1,5	1,5	ISO 6571

**Table 3 plants-10-01001-t003:** Effects of pre-harvesting conditions of *Origanum syriacum*.

Conditions	Main Effects	References
Irrigation level	Height and branching rate significantly increase with increasing of irrigation frequency.	[16]
	Thymol and carvacrol content of essential oil are not significantly affected.	[16]
Geographical location and cultivation method	Both essential oil yield and thymol and carvacrol levels are affected.	[19,20,21,22,23]
Harvesting period	The highest essential oil yield is obtained from summer harvesting.	[20,21,23,24]
	Thymol and carvacrol levels are higher in summer during the flowering period	[21,23]
	Wild *O. syriacum* leaves exhibit a higher level of thymol and carvacrol in June after flowering.	[25]
pH of soil	Significant positive correlation between essential oil yield and soil pH as well as soil phosphorus content.	[23]

**Table 4 plants-10-01001-t004:** Effect of post-harvesting conditions of *Origanum syriacum*.

Conditions	Main Effects	References
Drying temperature	Loss of essential oil by evaporating at drying temperatures higher than 30–35 °C.	[7]
	Drying at a lower temperature is better for preserving chlorophyll and carotenoids contents.	[18]
Air-drying	The best for preserving the aroma, flavor, and color compared to oven and freeze-drying.	[16,17]
Vacuum oven-drying	Higher levels of total phenols content and antioxidant capacity compared to freeze-drying.	[17]
Solar-drying	Decrease in chlorophyll content is more noticeable in drying at the high temperature of solar-drying. The loss of polyphenols is found to be the highest after drying at 26 °C and 0.1 m/s and lowest after solar-drying and drying at 45 °C and 0.2 m/s.	[26]
Essential oil extraction method	Hydro-distillation produces higher essential oil yield compared to supercritical fluid extraction.	[27]
	Static headspace (HS) exhibits better preservation of volatiles compared to steam distillation.	[20]
	Hydro-distillation shows better identification of essential oil components compared to solvent extraction.	[28]

**Table 5 plants-10-01001-t005:** Microorganisms tested with the *Origanum syriacum* essential oil and the relative references.

Microorganism	Reference
**Gram-Positive Bacteria**	
*Bacillus brevis*	[5]
*Bacillus cereus*	[42]
*Bacillus megaterium*	[5]
*Bacillus subtilis*	[5]
*Clostridium perfringens*	[42]
*Corynebacterium xerosis*	[5]
*Enterococcus faecalis*	[5,22]
*Enterococcus faecium*	[19]
*Listeria innocua*	[35]
*Micrococcus luteus*	[5]
*Mycobacterium smegmatis*	[5,42]
*Staphylococcus aureus*	[19,22,32] [5,42,43,44]
*Streptococcus pneumoniae*	[42,45]
**Gram-Negative Bacteria**	
*Acinetobacter lwoffii*	[42]
*Brucella melitensis*	[26]
*Enterobacter aerogenes*	[42]
*Escherichia coli*	[5,19,32,41,42,44]
*Klebsiella oxytoca*	[5]
*Klebsiella pneumoniae*	[5,42,44]
*Moraxella catarrhalis*	[42]
*Proteus mirabilis*	[26,42]
*Proteus vulgaris*	[44]
*Pseudomonas aeruginosa*	[5,19,26,29,42,44]
*Salmonella enterica*	[26]
*Salmonella typhi*	[44]
*Yersinia enterocolitica*	[5,26]
**Fungi**	
*Aspergillus flavus*	[22]
*Aspergillus fumigatus*	[22]
*Aspergillus niger*	[8,22]
*Candida albicans*	[32,42,44]
*Candida krusei*	[42]
*Fusarium oxysporum*	[8]
*Microsporum canis*	[44]
*Penicillium* species	[8]
*Saccharomyces cerevisiae*	[5]
*Trichophyton rubrum*	[32,44]

**Table 6 plants-10-01001-t006:** Antioxidant assays tested *Origanum syriacum*’s effect.

Antioxidant Assay	Reference
Ascorbate-Iron (III)-Catalyzed Phospholipid Peroxidation	[45]
DPPH Radical-Scavenging Activity	[5,19,31,35,36,42,45]
Ferric-Reducing Antioxidant Capacity (FRAC)	[35]
Ferrous Ion Chelating (FIC)	[35]
Iron (II) Chelation Activity	[45]
Iron (III) to Iron (II) Reducing Activity	[45]
Nonsite- and Site-Specific Hydroxyl Radical-Mediated 2-Deoxy-D-Ribose Degradation	[45]
RANCIMAT	[35]
Reducing Power Oyaizu Method	[5]
TBARS	[35]
Thiocyanate Method	[5]
TLC Plate	[42]
β-Carotene Bleaching Test	[31]
β-Carotene/Linoleic Acid	[42]

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
