# Peer review of "Origanum syriacum L. (Za’atar), from Raw to Go: A Review"

_plants, 2021, doi:10.3390/plants10051001_

Round 1

Reviewer 1 Report

The study is quite impressive and scientifically elaborated. I advise some corrections:

- I suggest for authors to add some section which are talking about others activities or potential of Za’atar (essential oils or extracts) such as anticancer (anti-tumoral) and anti-viral activities.

In the table 3: I suggest to add some information about the Origanum species that the study used (name, country of the collect, major bioactive compound,..).

In the Origanum distribution; I suggest adding a table that shows the names of each species of Origanum and its distribution ie the countries where it can be found.

- Correct some English mistakes before publication.

Author Response

Q. I suggest for authors to add some section which are talking about others activities or potential of Za’atar (essential oils or extracts) such as anticancer (anti-tumoral) and anti-viral activities.

R. This manuscript is addressed to a special issue dedicated to food plants so we think that other activities of za'atar essential oil, like anticancer or anti-viral, are out of the topic.

Q. In the table 3: I suggest to add some information about the Origanum species that the study used (name, country of the collect, major bioactive compound,..).

R. All the papers cited in the table 3 are just referred to the O. syriacum species as reported in the title of the table. About the country and bioactive compounds, unfortunately these are frequently missed information as we report in the text. Moreover, we report in the text the paper that found correlation between antimicrobial activity and specific components of the essential oil.

Q. In the Origanum distribution; I suggest adding a table that shows the names of each species of Origanum and its distribution ie the countries where it can be found.

R. We reported in the section 2.2 the distribution of Origanum species. We want to underline that this review is focused on Oryganum syriacum and we think that a table like that you suggest could be out of the topic.

Q. Correct some English mistakes before publication.

R. we checked for English.

Reviewer 2 Report

Dear Authors!

I marked in yellow in the manuscript my point of view.

Best regards.

MaPa

Author Response

Thank you so much for your help to make better the manuscript. We followed most of your suggestions. Just, unfortunately, we cannot add more citations for some parts of the review because we cannot find many papers about some aspects of this of product. We hope, with this review, to increase the interest of the scientific community toward Origanum syriacum and za'atar product.

Reviewer 3 Report

The present review deal with an important neglected and underutilized plant species such as origanum. The review is quite original however the main drawback that the review paper seems more suitable for a chapter book or an extension journal. I urge the authors to re-formulate the whole table of contents and make it more scientific. After a short rationale about the important of the species, the origin and the production area especially in the middle east, it is important to report the most important preharvest (irrigation, fertilization, growing seasons, planting date, landraces) and postharvest on yield and especially the quality of this important species in terms of essential oil and antioxidant compounds. The third part of the review should deal with the microbiological aspects.

I also urge the authors to report several tables at least 3 dealings with the preharvest, postharvest and microbiological aspects on the crop performance the resilience and quality of this species.

Author Response

We want to thank the reviewer for his comments and suggestions. Just we need to better understand what you mean to make the paper more scientific. It seems what you suggest is waht we report in the review. We added 2 new tables in which we describe pre-harvesting and post-harvesting factors can affect the product, as you suggest. However, it's not clear what you mean about microbiological aspects. Do you refer to the spoilage of za'atar and safety aspects? If yes, at best of our knowledge it's a missing topic. Just we published a previous paper (Journal of Food and Nutrition Research, 2020, Vol. 8, No. 6, 244-251) in which we underline the microbiological safety aspects of za'atar.

Round 2

Reviewer 1 Report

-No comments 

Reviewer 2 Report

Dear Authors!

I appreciate the manuscript and I accept this revised form.

Reviewer 3 Report

Dear authors

I am satisfied of the revised version and the addition of two important tables for the readers of Plants. Thanks for clarifying that a previous review was published on the microbiological aspects.

congrats!